# Plant Growth Promoting Bacteria and Arbuscular Mycorrhizae Improve the Growth of *Persea americana* var. Zutano under Salt Stress Conditions

**DOI:** 10.3390/jof9020233

**Published:** 2023-02-10

**Authors:** Richard Solórzano-Acosta, Marcia Toro, Doris Zúñiga-Dávila

**Affiliations:** 1Laboratorio de Ecología Microbiana y Biotecnología, Departamento de Biología, Facultad de Ciencias, Universidad Nacional Agraria La Molina, Lima 15024, Peru; 2Centro de Ecología Aplicada, Instituto de Zoología y Ecología Tropical, Facultad de Ciencias, Universidad Central de Venezuela, Caracas 1041-A, Venezuela

**Keywords:** salt stress, *Bacillus*, *Pseudomonas*, *Persea americana*, mycorrhizae

## Abstract

In Peru, almost 50% of the national agricultural products come from the coast, highlighting the production of avocado. Much of this area has saline soils. Beneficial microorganisms can favorably contribute to mitigating the effect of salinity on crops. Two trials were carried out with var. Zutano to evaluate the role of native rhizobacteria and two *Glomeromycota* fungi, one from a fallow (GFI) and the other from a saline soil (GWI), in mitigating salinity in avocado: (i) the effect of plant growth promoting rhizobacteria, and (ii) the effect of inoculation with mycorrhizal fungi on salt stress tolerance. Rhizobacteria *P. plecoglissicida*, and *B. subtilis* contributed to decrease the accumulation of chlorine, potassium and sodium in roots, compared to the uninoculated control, while contributing to the accumulation of potassium in the leaves. Mycorrhizae increased the accumulation of sodium, potassium, and chlorine ions in the leaves at a low saline level. GWI decreased the accumulation of sodium in the leaves compared to the control (1.5 g NaCl without mycorrhizae) and was more efficient than GFI in increasing the accumulation of potassium in leaves and reducing chlorine root accumulation. The beneficial microorganisms tested are promising in the mitigation of salt stress in avocado.

## 1. Introduction

Soil salinization is recognized as the main threat to environmental resources and affects between 7 and 10% of the continental extension [1]. A soil is considered saline when the ion concentration interferes with the growth of the species of agricultural interest and reaches an electrical conductivity >4 dS m^−1^ (approximately 36 mM NaCl), determined in the soil paste at 25° [2]. The presence of salts in the soil is of pedogenic (weathering) or anthropic (irrigation or fertilization water) origin and is exacerbated when rainfall is insufficient to leach ions from the soil profile, or when evaporation rates are high [2,3,4].

Salinity decreases agricultural production in most crops by affecting the physical and chemical properties of the soil and the ecological balance of the cultivated area [1,4]. High levels of salinity (>4 dS m^−1^) induce physiological stress because its ions produce toxicity, osmotic, and oxidative stress, and nutrient deficiency (N, Ca, K, P, Fe, Zn) which limit the uptake of nutrients from the soil. Salinity also affects photosynthesis, due to the reduction in leaf area, chlorophyll content, stomatal conductance and, to a lesser extent, the efficiency of photosystem II. This is reflected in a decrease in germination, vegetative growth, and reproductive development in different crops [2,3,5].

In this context, the avocado has been identified as one of the fruit trees most sensitive to salts [6,7], since the trees of this species irrigated with water whose chloride and sodium content is at tolerable levels for most of the crops show severe damage to the leaves, restriction of vegetative growth, and decreased production volume [6]. This situation occurs worldwide in those areas with an arid and semi-arid climate where avocados are grown [8], which has led to the search for different measures to deal with salinity, among which the selection of salt-tolerant rootstocks represents a feasible but not sufficient alternative [9].

On the desert coast of Peru, a pole of agro-industrial development and avocado cultivation has been established since 2018 as the main agro-export sector [10]; there are approximately 33,000 hectares of “Hass” avocado [11], where the degradation of these lands is mainly due to soil salinity, which affects approximately 40% of the arable surface in the Piura, Tumbes, and Lambayeque regions, which are the most important areas for agricultural production for export [12].

In addition, it has been determined that in the coastal areas of Peru, there are approximately 300,000 hectares that have salinization problems, which not only affect avocado cultivation, but also, and especially, areas where rice is grown [13] among others. These saline soils have been formed in part by natural conditions such as rainfall deficit, arid and semi-arid environments; although, in most cases, it has increased due to artificial conditions such as irrigation with saline waters, inadequate irrigation, and poor drainage [14]. So, saline soils in Peru are widely distributed and are generally treated by amendments with calcium sulfate, application of organic amendments, and reseeding or change of crop, whose action is local and temporary and even expensive. Therefore, methods are required that are long-lasting and sustainable to the environment, while maintaining agricultural activities [15].

Soil salinity then generates a depressing effect on plant growth, and some plant growth promoting bacteria (PGPR) or other microorganisms that can attenuate the stress that this produces [16]. In the case of the genus *Bacillus*, various authors indicate that *B. subtilis* reduces the level of oxidative and osmotic stress induced in seedlings that manifests itself in the improvement of plant growth [17,18,19]. In the same way, the genus *Pseudomonas* has been shown to reduce the level of salinity stress in cultivated plants by decreasing peroxidase activity in the plant and increasing nutrient uptake and improving plant growth under salinity conditions at high concentrations [20,21].

On the other hand, arbuscular mycorrhizae (AM) associate with most plants of agricultural interest and improve growth under stress conditions due to adverse soil conditions; specifically, they modulate salinity stress [22,23] by root colonization, modifying antioxidant enzyme systems that reduce oxidative damage and stimulate growth [24,25,26]. AM symbiosis increases proline synthesis and superoxide dismutase, catalase, and ascorbate peroxidase activities that decrease the synthesis of phenolic compounds related to salinity stress [27].

Regarding the joint use of mycorrhizae and plant growth-promoting bacteria (PGPR), it is known that they can influence the superior response of plants to abiotic stresses such as drought and salinity through different mechanisms such as increased potassium uptake and decreased sodium by the plant [28,29]. In addition, synergistic effects potentially contribute to expanding crop production to soils that would otherwise be unproductive through interactions that may have a potential role in relieving stress [29], or improving nutrient uptake and crop productivity cultures compared to a single inoculation [30,31].

Therefore, microorganisms that promote plant growth constitute potential alternatives for sustainable agriculture due to their low cost of production and high effectiveness. Hence, the objective of this research was to evaluate PGPR and arbuscular mycorrhizae in improving resistance to salinity stress in avocado seedlings of the Zutano variety.

## 2. Materials and Methods

### 2.1. Selection of PGPR Resistant to High Osmotic Concentrations

From 26 isolates of bacteria belonging to the genera *Bacillus* and *Pseudomonas* from the rhizosphere of *P. americana*, 7 strains were selected according to the growth-promoting activity associated with phosphate solubilization [32], synthesis of siderophores [33], and antagonistic activity against *P. cinnamomi* and/or *L. theobromae* for representing the most important pathogens for avocado cultivation in Peru and the world. Then, those showing greater growth under salinity conditions in increasing concentrations of salinity in nutrient agar with 7.5 and 15 g·L^−1^ of sodium chloride, respectively, were selected.

Previously, the strains isolated from the rhizosphere were identified by amplifying the 16S rDNA gene, for which they were activated in nutrient broth at 37 °C for 18 h, and the genomic DNA of each strain was extracted according to the instructions of the commercial kit (Thermo SCIENTIFIC, USA) and later sequenced by MacroGen Inc. (Seoul, Korea). To determine their taxonomic identity, BLAST analysis of the similarity between the sequences of the nucleotides obtained and those deposited in the Gen Bank database was used. Phylogenetic trees of 16S rRNA were deduced using the neighbor-joining analysis [34]. Evolutionary analyses were performed in MEGA X (2018).

### 2.2. Description of the Substrate Used for the Tests

Premix ® N°8 substrate (pH: 5.5, EC: 0.75 dS m^−1^, P: 25 ppm and K: 100 ppm) was added to 2-L plastic pots; one avocado seed per pot was sown of the var. Zutano weighing approximately 50 g; the seeds were donated by the company Avo Hass Peru, previously sterilized in 10% sodium hypochlorite for 10 min and once rinsed, each one of them was allowed to germinate for ten days.

### 2.3. Preparation of Avocado Seeds for the Installation of the Experiment

Once the seeds germinated after ten days in the 2 L pots, they were inoculated with the bacteria or *Glomeromycota fungi* according to the corresponding test and they were allowed to grow for three weeks in a greenhouse with an average temperature of 20ºC and humidity of 60% until the seedlings were obtained, prior to treatment with irrigation with saline water.

### 2.4. Selection, Propagation, and Inoculation of Arbuscular Mycorrhizae

Two inocula of *Glomeromycota fungi* isolated from Peru were used. One of them was obtained from rhizospheric soil from a fallow land in Pucallpa, Ucayali (Glomeromycota Fallow Inoculum-GFI) and the other one from rhizospheric soil of *Sporobolus* sp. located in Pisco, Ica (Glomeromycota Wetland Inoculum-GWI), according to Castañeda et al. [35]. Both were reproduced in trap pots with *Brachiaria decumbens* in sterile sand, irrigated with Long Ashton’s solution, and a quarter of the dose of P every 15 days [36]. Rhizospheric samples were taken to quantify spores and AM colonization of the rootlets [37,38] in each inoculum. A similar number of spores was obtained in GFI and GWI to inoculate the avocado plants in the experiment. The GFI inoculum consisted of the native species, mainly the fungus *R. intraradices* and the genera *Acaulospora*, *Gigaspora*, and *Archaeospora*, of which 3726 spores/mL per seedling were added from colonized rootlets (70%). In the GWI inoculum, the fungus *R. intraradices* predominated, of which 3400 spores/mL per seedling were applied from colonized rootlets (90%). In both cases, the inocula were applied at the base of the pre-germinated seeds, spreading the inoculum around, slightly uncovering the substrate with a sterile spatula to expose the rootlets without damaging them.

### 2.5. Inoculation of the Seedlings with the Bacterial Strains

The bacteria selected for their resistance to growing in saline conditions in vitro were cultured for 48 h in nutrient broth at 28 °C; then, 30 mL of bacterial broth from each strain with a concentration of 10^8^ cfu mL^−1^ was applied to the neck of the plant, 15 days after germination of the seed, and the area of influence of the rootlets. This procedure was repeated a second time 3 days after the first inoculation to guarantee its presence and adequacy in the rhizosphere of the plant before starting the saline irrigation treatments.

### 2.6. Greenhouse Environmental Conditions

Throughout the experiment, the maximum temperatures recorded varied between 24.51 and 27.43 °C, and the minimum between 18.91 and 20.37 °C. The relative humidity varied between 59.9 and 61.92%. These parameters were recorded by the meteorological station located inside the greenhouse of the Microbial Ecology and Biotechnology Laboratory of the La Molina National Agrarian University.

### 2.7. Effect of Plant Growth Promoting Rhizobacteria on Salt Stress Tolerance in Persea americana var. Zutano

Nine treatments were established based on two factors: (a) inoculation of plant growth promoting rhizobacteria (*Bacillus subtilis*, *Pseudomonas plecoglossicida* and without bacteria) and (b) sodium chloride concentrations (irrigation water without sodium chloride, irrigation water with 0.75 g/L of sodium chloride, and irrigation water with 1.5 g/L of sodium chloride). The test was maintained for 60 days with an irrigation frequency of three times a week and a dose of 500 mL of water per plant.

### 2.8. Effect of Inoculation with Glomeromycota fungi on Tolerance to Salt Stress in Persea americana var. Zutano

Seven treatments were established based on two factors: (a) mycorrhizae (isolated from wetland soils, GWI, and isolated from fallow soils, GFI) and (b) sodium chloride concentrations (irrigation water without sodium chloride, irrigation water with 0.75 g/L of sodium chloride and irrigation water with 1.5 g/L of sodium chloride). The test was maintained for 60 days with an irrigation frequency of three times a week and a dose of 500 mL per plant.

### 2.9. Evaluated Parameters

Growth parameters were evaluated in the seedlings associated with plant height, stem diameter, and the accumulation of fresh and dry matter both in the root and in the aerial part, for which the samples were dried at 40 °C for 5 days until no variation in the weight of the sample was observed. Additionally, the content of ions related to saline stress, such as K, Na, and Cl, was measured both in the aerial biomass (leaves and stems) and in the root biomass.

### 2.10. Statistical Analysis

For the statistical analysis, the Statistic Package for Social Sciences (SPSS) program from IBM company version 26 was used. The data obtained in each experiment were submitted to the analysis of variance for a complete randomized design (DCA) with factorial arrangement using the F test and by calculating the main effects of the tested factors; then, the treatments were compared in case of significance using Duncan’s test to determine differences between strains and salinity levels tested. The probability of alpha error of less than 5% was considered significant.

## 3. Results

### 3.1. Selection of Plant Growth Promoting Rhizobacteria Tolerant to High Salinity Concentrations

Table 1 shows the seven potential strains and their growth-promoting capacity; then, Figure 1 and Figure 2 show their phylogenetic relationship with respect to the 26 initial isolates determined in the avocado rhizosphere. Finally, of the seven potential strains with growth promoting capacity, the Bac F (*Bacillus subtilis*) and Bac M (*Pseudomonas plecoglossicida*) strains were tolerant to salinity because they showed growth slightly like the control medium (without NaCl) (Figure 3). A slight increase (4 mm) in the Bac M colony is noted with respect to the control and the proliferation of exudates in the Bac F strain that also increased its colony diameter (6 mm).

The phosphate solubilization index in NBRIP solid medium (1999) expressed as a percentage of the solubilization halo. Siderophore production is indicated by the color change from blue to yellow around bacterial growth. The effect on growth is reported in the ability to accumulate dry biomass in the root and aerial part expressed in grams. In vitro antagonistic activity expressed as the percentage of growth inhibition at 72 h, which was calculated by inhibition halo.

### 3.2. Effect of Plant Growth Promoting Rhizobacteria on Salt Stress Tolerance in Persea americana var. Zutano

The main effects were calculated for each factor tested within the factorial arrangement (salinity and type of PGPR bacteria) by means of analysis of variance (Table 2). The inoculation of *B. subtilis* stimulated the height of the plant (Table 2) and, in general, favored the appearance of the plant (coloration and vigor) (Figure 4). It also increased the dry root biomass (Table 2) in the levels of salinity tested. Regarding salinity, as it increases, avocado growth decreases (plant height, stem diameter, fresh aerial biomass, and dry aerial biomass).

### 3.3. Effect of Plant Growth Promoting Rhizobacteria on the Accumulation of Ions Associated with Salinity in Persea americana var. Zutano

In the leaves, first *P. plecoglossicida* and then *B. subtilis* decrease the accumulation of chlorine and sodium ions, while they increase the accumulation of potassium (Table 3). On the other hand, in the roots, the order is inverted;. *B. subtilis* is the bacterium that allows the reduction of chlorine and sodium ions, and increases the accumulation of potassium, and it is followed by the bacterium *P. plecolossicida* with the same effects compared to the treatments where bacteria are not inoculated, and where salinity levels increase (Table 4).

### 3.4. Effect of Inoculation with Glomeromycota fungi (GFI and GWI) on Tolerance to Salt Stress in Persea americana var. Zutano

Regarding growth parameters, inoculation with arbuscular mycorrhizae (GFI, GWI) increased plant length, aerial fresh biomass, and root dry biomass, in addition to stabilizing other growth parameters in the presence of increasing salinity levels (Table 5, Figure 5 and Figure 6).

Regarding the uptake of saline ions, in leaves, there is a tendency for AM to facilitate the accumulation of sodium, potassium, and chlorine ions at a low level of salinity, probably because they improve nutrient absorption. This increase is not substantial and ends up decreasing at a higher level of salinity, lessening the impact of salinity understood as the accumulation of these ions below control where *Glomeromycota fungi* were not applied (Table 6). The effect of mycorrhizae is clearer in the roots. Regarding sodium, the inoculum from a wetland soil of the Peruvian coast (GWI) decreased its accumulation compared to the control (1.5 g NaCl without mycorrhizae). GWI was more efficient than that from the fallow (GFI) in increasing potassium accumulation compared to the control (1.5 g NaCl without mycorrhizae). Moreover, GWI inoculation favored the decrease in chlorine accumulation compared to the control (1.5 g NaCl without mycorrhizae) (Table 7).

As can be seen in Figure 6, the absolute control without salinity or mycorrhizae is a healthy and vigorous plant (G). The treatments with increasing salinity (E and F) without mycorrhizae, according to the level of salinity, dry and defoliate causing the death of the seedling. Inoculation with mycorrhizae is more effective at a concentration of 0.75 g/L of NaCl (A and C).

## 4. Discussion

### 4.1. Effect of Plant Growth Promoting Rhizobacteria on Salt Stress Tolerance in Persea americana var. Zutano

In general, saline stress is the main restrictive factor in the production of horticultural crops [39,40]. The presence of *Bacillus subtilis* in the rhizosphere of the seedlings modified their tolerance to stress induced with NaCl (Figure 4). It is believed that these responses are related to the ability of the bacterium (*B. subtilis*) to attenuate the synthesis of reactive oxygen species [18,19]. In the plant, research has shown that PGPR bacteria can help mitigate the adverse effects of salinity [41] by various mechanisms. Several PGPRs possess acdS genes that encode ACC deaminase and enhance plant growth and development by decreasing ethylene synthesized due to salinity, since deaminases reduce ethylene production by converting ACC to α-ketobutyrate and ammonia [42]. ACCD hydrolyzes ACC (precursor of ethylene biosynthesis in higher plants) into alkali and α-ketobutyrate for use as a nitrogen source and enhances plant growth under saline conditions [43]. As ACC deaminase works by reducing ACC levels within the plant, the inhibition of plant growth and development by ethylene (particularly in conditions of stress, including salinity stress) is decreased, and these plants, in most cases, have longer roots and shoots and greater biomass [44] as was obtained in the present investigation (Table 2). The PGPR also produce IAA, which is absorbed by the plant and promotes the expansion and elongation of plant cells. They can also produce abscisic acid (ABA) involved in the water economy of the plant and stomatal control [45], both parameters affected by saline stress.

### 4.2. Effect of Plant Growth Promoting Rhizobacteria on the Accumulation of Ions Associated with Salinity in Persea americana var. zutano

In the leaves, first *P. plecoglossicida* and then *B. subtilis* decrease the accumulation of chlorine and sodium ions, while they increase the accumulation of potassium (Table 3). On the other hand, in the roots, the order is inverted. *B. subtilis* is the bacterium that allows the reduction of chlorine and sodium ions, and increases the accumulation of potassium, followed by the bacterium *P. plecolossicida* with the same effects compared to the treatments where bacteria are not inoculated, and where salinity levels increase (Table 4).

As can be seen, the PGPR of the genera *Bacillus* and *Pseudomonas* decrease the chlorine and sodium content in plant tissues in avocado seedlings exposed to soil salinity; while, they increase the potassium content associated with water management by the plant under salinity conditions. This effect on ions related to salinity in plant physiology has been demonstrated in recent years by various authors, especially on the bacteria genera tested. This is what [46] points out about the cultivation of maize when soaking the seeds in a *Pseudomonas* solution in a trial in which they managed to reduce the absorption of Na in the vegetable by up to 50%. In addition, [47] demonstrated that the application of *P. geniculate* reduces Na^+^ uptake and increases K^+^ and Ca^2+^ uptake in maize roots, which highlights the role of PGPR bacteria in maintaining ionic balance in plant roots under saline conditions.

In avocado cultivation, experiences are less precise with respect to ions but show that the application of bacterial consortia improve avocado growth under salinity conditions as well as reduce physiological damage caused by it [48]. In general, inoculation with PGPR not only helps to alleviate the toxic effects of salt, but also increases plant growth along with reducing crop losses due to salinity and sodicity [49,50].

### 4.3. Effect of Inoculation with Glomeromycota fungi (GFI and GWI) on Tolerance to Salt Stress in Persea americana var. Zutano

Fresh or dry biomass is the characteristic most affected by salinity and where mycorrhizae have had the most effect (Table 5). In general, some studies in recent years have reported that the use of native mycorrhizae strains has been effective as an inoculum to enhance the development of avocado seedlings [51,52,53]. Similar effects regarding potassium are reported by [54] in mycorrhized plants.

It is known that high Na concentration negatively interferes with transporters located in the root plasma membrane, such as K^+^ selective ion channels [55]. As a result, the absorption of some mineral nutrients (N, P, K, Fe, Cu, and Zn) is reduced. The high Na^+^/K^+^ ratio interrupts various enzymatic processes and protein synthesis [56]. AM and some PGPRs have been shown to enhance K^+^ uptake, helping plants to maintain a lower Na^+^/K^+^ ratio and a better ionic balance by favoring N, P, K, Cu, Fe, and Zn, thus preventing damage to normal cellular enzymatic processes [31]. AM can regulate the movement of Na^+^ ions excess from cells via the plasma membrane antiporter Na^+^/H^+^ by modulating salt sensitive (SOS) genes, thus maintaining ionic homeostasis [57]. The responses observed in mycorrhized plants are related to the regulation of the expression of plant genes involved in proline biosynthesis, which code for aquaporins. The regulation of these genes allows mycorrhizal plants to maintain a better water status in their tissues [54], which agrees with Sagar et al. [30] who mention that mycorrhizae help in the efficient absorption of mineral nutrients and water by plants, as well as enhance the absorption of nutrients to improve the tolerance of plants against salinity stress, which can be an extremely useful approach to sustainable agriculture.

## 5. Conclusions

The inoculation of seedlings with both PGPR, *P. plecoglossicida*, and *B. subtilis,* contributes to the reduction in the accumulation of chlorine and sodium ions, while they increase the accumulation of potassium in avocado leaves. Chlorine and sodium ions also decrease while potassium accumulation in leaves and roots increases.

The inoculation of seedlings with GFI and GWI *Glomeromycota fungi* reduced the impact of salinity, improving plant growth, and decreasing the accumulation of chlorine and sodium ions in leaves and roots below control where mycorrhizae were not applied. The beneficial microorganisms tested are promising in the mitigation of salt stress in avocado, highlighting the inoculum native from saline soil (GWI).

In the future, it is necessary to study the feasibility of using the co-inoculation of *Glomeromycota fungi* and plant growth promoting rhizobacteria from saline soils in avocado seedlings, particularly with both bacteria used in this work and GWI Glomeromycota inoculum. In addition, efficacy tests should be conducted in the field to allow the development of commercial inoculants with these microorganisms for the treatment of saline soils in avocado cultivation.

## Figures and Tables

**Figure 1 jof-09-00233-f001:**
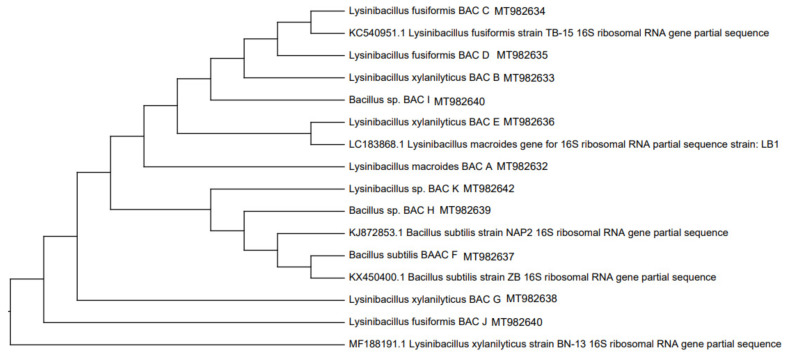
Phylogenetic tree constructed according to the maximum likelihood method based on the relationship between the 16S rRNA gene sequences of *Bacillus strains* isolated from the rhizosphere of *P. americana* and related *Bacillus* species. Sequences of the 16S rRNA gene of the *Bacillus subtilis* strain were arbitrarily chosen as a sequence outside the group.

**Figure 2 jof-09-00233-f002:**
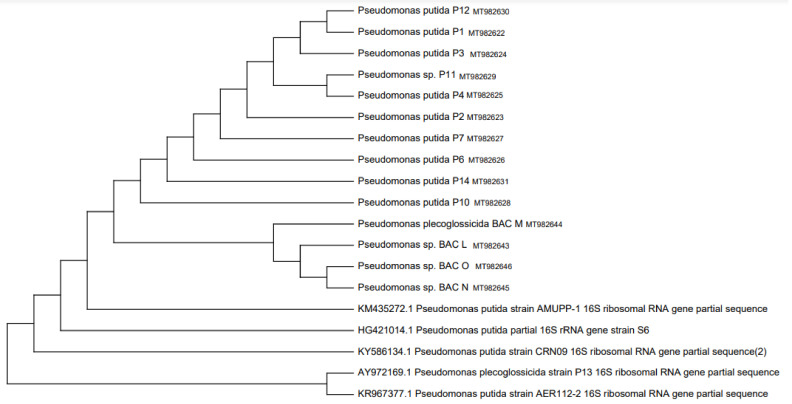
Phylogenetic tree built according to the maximum likelihood method based on the relationship between the 16S rRNA gene sequences of the *Pseudomonas* strains isolated from the *Persea americana* rhizosphere and related *Pseudomonas* species. Sequences of the 16S rRNA gene of the Pseudomonas putida strain was arbitrarily chosen as a sequence outside the group.

**Figure 3 jof-09-00233-f003:**
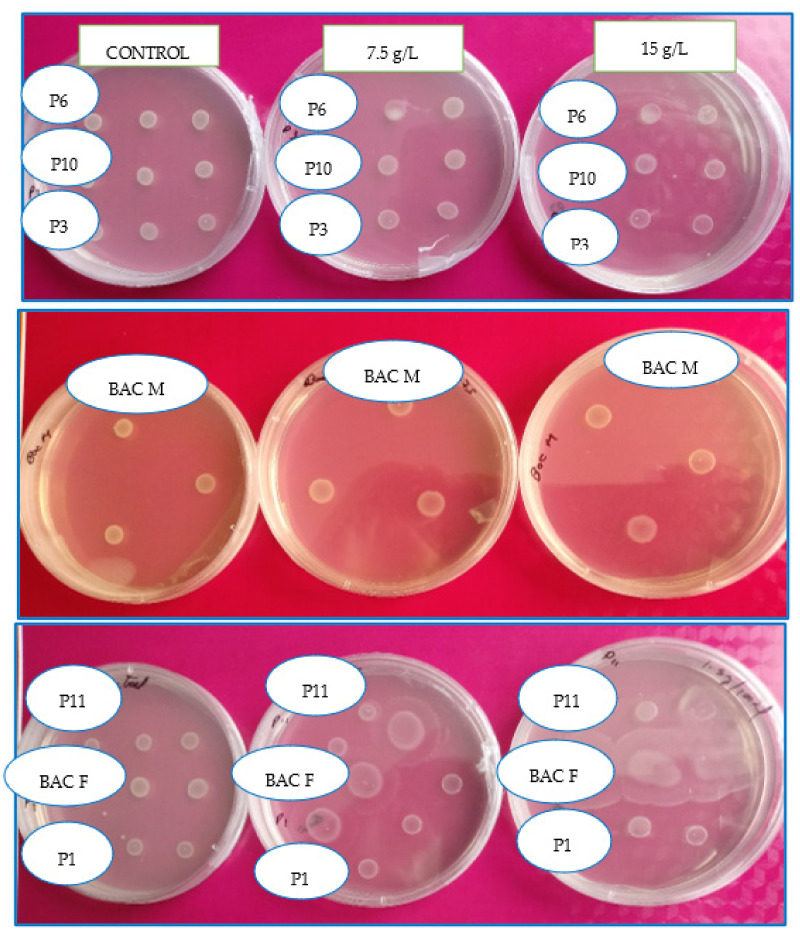
Growth of avocado rhizospheric bacteria in media with increasing concentrations of NaCl.

**Figure 4 jof-09-00233-f004:**
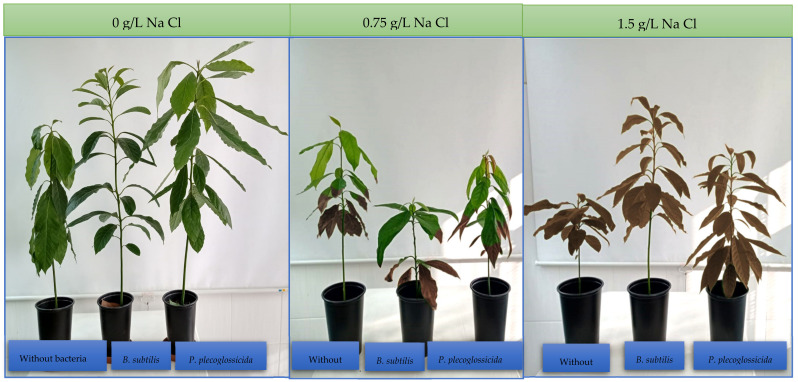
Effect of the inoculation of PGPR on the growth of *Persea americana* var. Zutano under salinity stress conditions.

**Figure 5 jof-09-00233-f005:**
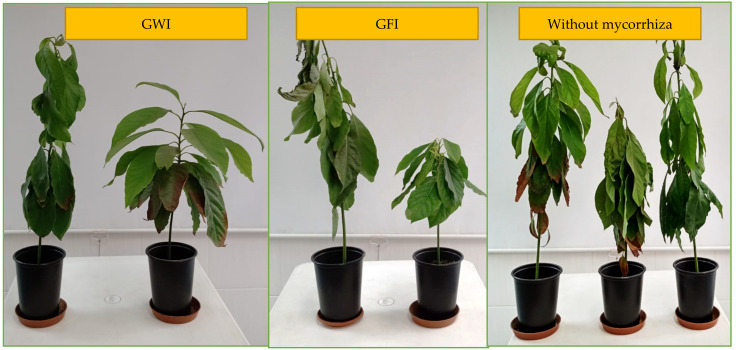
Effect of inoculation with arbuscular mycorrhizae on *Persea americana* var. Zutano under salinity stress conditions.

**Figure 6 jof-09-00233-f006:**
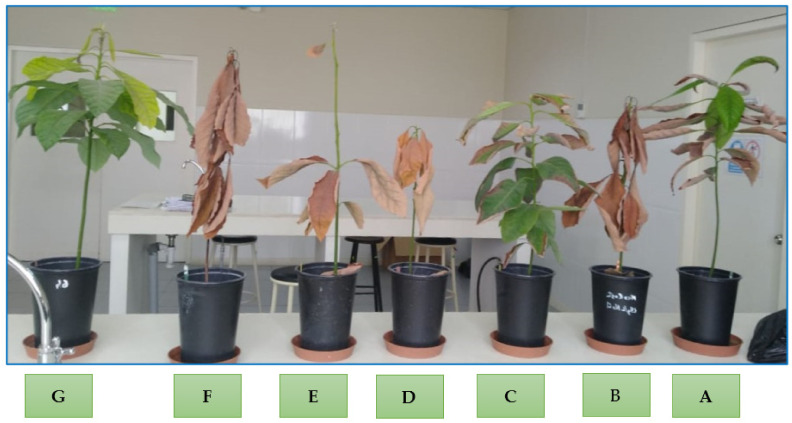
Seedlings of *Persea americana* var. Zutano according to the following treatments: (**A**) GWI + 0.75 g/L NaCl, (**B**) GWI + 1.5 g/L NaCl, (**C**) GFI+ 0.75 g/L NaCl, (**D**) GFI + 1.5 g/L NaCl, (**E**) without mycorrhizae + 0.75 g/L NaCl, (**F**) without mycorrhizae + 1.5 gLl NaCl, (**G**) without mycorrhizae/without NaCl.

**Table 1 jof-09-00233-t001:** PGPR activity of selected *Bacillus* and *Pseudomonas* strains.

Características	Strain						
	P1	P3	P6	P10	P11	BAC F	BAC M
-Identity (BLAST NCBI)	*Pseudomonas putida*	*Pseudomonas putida*	*Pseudomonas putida*	*Pseudomonas putida*	*Pseudomonas sp.*	*Bacillus subtilis*	*Pseudomonas plecoglossicida*
-Phosphate solubilization capcity (%)	200	100	0	100	183	0	266
-Siderophore production	-	-	-	+	-	-	-
-Root biomass production(g)	2.6	1.6	2.4	1.7	1.9	3.3	2.8
-Leaves biomass production (g)	2.7	2.2	2.8	1.8	2.4	4.3	3.8
-Antagonistic activity on *P. cinnamomi* (%)	40	48	47	42	40	45	28
-Antagonistic activity on *L. theobromae* (%)	0	0	0	0	0	33	0

**Table 2 jof-09-00233-t002:** Growth parameters of *Persea americana* var. Zutano inoculated with PGP bacteria under salinity stress conditions.

Treatment		Plant Height (cm)	Stem Diameter (mm)	Fresh Aerial Biomass (g)	Dry Aerial Biomass (g)	Fresh Root Biomass (g)	Dry Root Biomass (g)	Total Dry Biomass (g)	Total Dry Biomass Decrease (%) *
Without NaCl	*B. subtilis*	73.9 f	0.68 a	57.16 d	27.76 d	20.51 cd	16.33 cd	44.09 e	0.00
0.75 g/L NaCl	*B. subtilis*	47.20 b	0.63 a	29.05 a	16.34 bc	17.16 bc	17.22 d	33.56 d	23.88
1.5 g/L NaCl	*B. subtilis*	58.80 d	0.63 a	22.34 a	11.48 a	15.30 ab	5.64 a	17.12 a	61.17
Without NaCl	*P. plecoglossicida*	64.51 e	0.65 a	68.63 f	26.12 d	27.54 e	13.99 bcd	40.11 d	0.00
0.75 g/L NaCl	*P. plecoglossicida*	50.30 c	0.62 a	27.83 c	13.75 bc	24.17 de	12.18 bc	25.93 b	35.35
1.5 g/L NaCl	*P. plecoglossicida*	45.15 b	0.51 a	18.91 a	9.56 a	12.12 a	3.74 a	13.3 a	66.84
Without NaCl	Without bacteria	63.20 e	0.63 a	63.07 e	28.41 d	35.76 f	18.19 d	46.6 d	0.00
0.75 g/L NaCl	Without bacteria	58.70 d	0.60 a	21.4 a	17.84 c	19.96 bcd	12.38 bc	30.22 bc	35.15
1.5 g/L NaCl	Without bacteria	42.15 a	0.54 a	27.06 bc	11.90 a	19.31 bcd	9.82 b	21.72 b	53.39

Equal letters indicate statistically equal averages from Duncan’s Test at 95% confidence. * Expresses the decrease in total dry matter (MST) expressed as a percentage grouped by each bacterial species (*B. subtillis*, *P. plecoglossicida* and Without bacteria) as the saline concentration increases.

**Table 3 jof-09-00233-t003:** Accumulation of saline ions in leaves of *Persea americana* var. Zutano inoculated with PGPR and induced to salinity stress.

Salinity	Bacteria	K	Na	Cl
Without NaCl	*B. subtilis*	1.02 a	0.078 a	0.24 a
	*P. plecoglossicida*	1.18 c	0.008 a	0.42 c
	Without bacteria	1.11 b	0.010 a	0.37 b
0.75 g NaCl	*B. subtilis*	1.70 g	0.089 b	3.66 e
	*P. plecoglossicida*	1.69 g	0.010 c	3.98 f
	Without bacteria	1.86 h	0.097 bc	3.48 d
1.5 g NaCl	*B. subtilis*	1.63 f	0.27 e	6.04 h
	*P. plecoglossicida*	1.59 e	0.12 d	4.67 g
	Without bacteria	1.38 d	0.37 f	6.28 i

Equal letters in the same column indicate statistically equal means from Duncan’s Test at 95% confidence.

**Table 4 jof-09-00233-t004:** Accumulation of saline ions in leaves of *Persea americana* var. Zutano inoculated with PGPR under salinity stress conditions.

Salinity	Bacteria	K	Na	Cl
Without NaCl	*B. subtilis*	1.26 h	0.05 a	0.85 b
	*P. plecoglossicida*	1.21 g	0.04 a	0.82 a
	Without bacteria	1.15 f	0.04 a	0.95 c
0.75 g NaCl	*B. subtilis*	0.66 d	0.22 d	0.86 b
	*P. plecoglossicida*	0.71 e	0.15 b	1.76 d
	Without bacteria	0.44 c	0.14 b	3.37 h
1.5 g NaCl	*B. subtilis*	0.41 b	0.21 d	1.93 e
	*P. plecoglossicida*	0.38 a	0.18 c	2.25 f
	Without bacteria	0.45 c	0.30 e	2.52 g

Equal letters in the same column indicate statistically equal means from Duncan’s Test at 95% confidence.

**Table 5 jof-09-00233-t005:** Effect of inoculation with *Glomeromycota fungi* (GFI and GWI) on growth parameters in seedlings of *Persea americana* var. Zutano under salinity stress conditions.

Treatment	Plant Height (cm)	Fresh Aerial Biomass (g)	Fresh Root Biomass (g)	Dry Aerial Biomass (g)	Dry Root Biomass (g)	Total Dry Biomass (g)	Total Dry Biomass Increase (%) *
Without mycorrhiza/Without NaCl	70.87 d	47.33 d	31.88 d	26.54 e	10.12 c	36.66 e	-
Without mycorrhiza + 0.75 g/L NaCl	64.10 cd	24.13 b	22.95 c	15.15 bc	6.12 b	21.27 c	0
GFI + 0.75 g/L NaCl	60.5 cd	40.03 cd	15.78 ab	20.14 cd	4.15 a	24.29 cd	14.19
GWI + 0.75 g/L NaCl	57.32 bc	36.00 c	19.92 bc	21.62 de	4.76 ab	26.38 d	24.02
Without mycorrhiza + 1.5 gLl NaCl	63.87 cd	19.55 ab	15.06 ab	12.54 ab	4.76 ab	17.3 b	0
GWI + 1.5 g/L NaCl	44.9 a	13.99 a	13.21 a	9.23 a	3.80 a	13.03 a	−24.68
GFI + 1.5 g/L NaCl	48.87 ab	13.03 a	14.80 ab	7.79 a	3.24 a	11.3 a	−34.68

Equal letters indicate statistically equal averages from Duncan’s Test at 95% confidence. GFI (jungle). GWI (coast). * Expresses the increase in total dry matter (MST) expressed as a grouped percentage for each saline concentration (0.75 g/L NaCl) for each type of *Glomeromycota fungi* inoculum tested.

**Table 6 jof-09-00233-t006:** Accumulation of saline ions in leaves of *Persea americana* var. Zutano inoculated with *Glomeromycota fungi* (GFI and GWI) under salinity stress conditions.

Treatments		Na	K	Cl
Without mycorrhiza	Without NaCl	0.08 a	1.2 a	0.50 a
	0.75 g NaCl	0.60 b	1.80 f	3.73 b
	1.5 g NaCl	3.00 g	1.67 c	6.32 g
GFI	0.75 g NaCl	0.62 c	1.72 d	3.94 c
	1.5 g NaCl	0.80 d	1.63 b	4.00 d
GWI	0.75 g NaCl	1.04 e	1.86 g	4.10 e
	1.5 g NaCl	2.38 f	1.74 e	5.11 f

Equal letters in the same column indicate statistically equal means from Duncan’s Test at 95% confidence.

**Table 7 jof-09-00233-t007:** Accumulation of saline ions in roots of *Persea americana* var. Zutano inoculated with *Glomeromycota fungi* (GFI and GWI) under salinity stress conditions.

Treatments		Na	K	Cl
Without micorriza	Without NaCl	0.34 a	0.96 f	0.62 a
	0.75 g NaCl	1.38 d	0.40 b	1.49 c
	1.5 g NaCl	1.34 c	0.28 a	2.49 f
GFI	0.75 g NaCl	1.60 e	0.48 d	1.65 d
	1.5 g NaCl	2.00 f	0.42 c	2.89 g
GWI	0.75 g NaCl	0.87 b	0.43 c	1.12 b
	1.5 g NaCl	1.34 c	0.50 e	2.27 e

Equal letters in the same column indicate statistically equal means from Duncan’s Test at 95% confidence.

## Data Availability

Not applicable.

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
