# Peer review of "Plant Growth Promoting Bacteria and Arbuscular Mycorrhizae Improve the Growth of Persea americana var. Zutano under Salt Stress Conditions"

_jof, 2023, doi:10.3390/jof9020233_

Round 1

Reviewer 1 Report

The Article entitled" Plant growth promoting bacteria and arbuscular mycorrhizae improve the growth of Persea americana var. Zutano under salt 3 stress conditions presents useful information data on plant cultivation and associated rhizozpheric microorganisms.

The manuscript is clearly written; however, some details can be improved. See below

To use the term: Mycorrhizae along the text

Line96: ...and high? effectiveness

line 177: jungle can be replaced by forest or some more precise term.

hours: replace by h

line 251, 409: showed to increase. Please, check this sentence (showed an increased....)

line405: PGP/ REPLACE BY PGPR

line 409: on the genera tested/ bacteria or plant?

line416: soil? ions

Conclusions:

AMF decrease...

to include the term: inoculation of seedlings

to indicate possible future research required.

o indicate possible future research required. Also, if is possibly to indicate which mycorrhizae could be used for inoculation.

Author Response

Reviewer 1.

Report Form

Open Review

 ( ) English very difficult to understand/incomprehensible
( ) Extensive editing of English language and style required
(x) Moderate English changes required
( ) English language and style are fine/minor spell check required
( ) I don't feel qualified to judge about the English language and style

English language and style

Yes

Can be improved

Must be improved

Not applicable

Does the introduction provide sufficient background and include all relevant references?

(x)

( )

( )

( )

Are all the cited references relevant to the research?

(x)

( )

( )

( )

Is the research design appropriate?

(x)

( )

( )

( )

Are the methods adequately described?

(x)

( )

( )

( )

Are the results clearly presented?

(x)

( )

( )

( )

Are the conclusions supported by the results?

( )

(x)

( )

( )

Comments and Suggestions for Authors

The Article entitled" Plant growth promoting bacteria and arbuscular mycorrhizae improve the growth of Persea americana var. Zutano under salt 3 stress conditions presents useful information data on plant cultivation and associated rhizozpheric microorganisms.

The manuscript is clearly written; however, some details can be improved. See below

  • To use the term: Mycorrhizae along the text.

Reply: The term was standardized throughout the text

  • Line96: ...and high? Effectiveness

Reply: The term high was added

  • line 177: jungle can be replaced by forest or some more precise term.

Reply: fixed by adding the term forest

  • hours: replace by h

Reply: Reviewed and changes were made throughout the text.

  • line 251, 409: showed to increase. Please, check this sentence (showed an increased....)

Reply: Suggested change was made

  • line405: PGP/ REPLACE BY PGPR

Reply: Reviewed and changes were made throughout the text.

  • line 409: on the genera tested/ bacteria or plant?

Reply: It was corrected indicating that it refers to the genus of bacteria: "bacteria generates tested"

  • line416: soil? Ions

Reply: It was changed to: "ionic balance in plant roots under saline conditions"

Conclusions:

  • AMF decrease...

Reply: It was specified by directly mentioning the inocula: GFI and GWI mycorrhizae

  • To include the term: inoculation of seedlings

Reply: The term seedling inoculation was added to be more specific in the conclusions

  • To indicate possible future research required, or indicate possible future research required. Also, if is possibly to indicate which mycorrhizae could be used for inoculation.

Reply: A paragraph with these aspects was added in conclusions. As follows:

The inoculation of seedlings with both PGPR, P. plecoglossicida and B. subtilis, contributes to the reduction of the accumulation of chlorine and sodium ions, while they increase the accumulation of potassium in avocado leaves. Chlorine and sodium ions also decrease while potassium accumulation in leaves and roots increases.

The inoculation of seedlings with GFI and GWI Glomeromycota fungi reduced the impact of salinity, improving plant growth and decreasing the accumulation of chlorine and sodium ions in leaves and roots below control, where mycorrhizae were not applied. The beneficial microorganisms tested are promising in the mitigation of salt stress in avocado, highlighting the inoculum native from saline soil (GWI).

In the future, it is necessary to study the feasibility of using the co-inoculation of Glomeromycota fungi and Plant Growth Promoting Rhizobacteria from saline soils in avocado seedlings, particularly with both bacteria used in this work and GWI Glomeromycota inoculum. Also conduct efficacy tests in the field that allow the development of commercial inoculants with these microorganisms, for the treatment of saline soils in avocado cultivation.

Submission Date: 27 December 2022

Date of this review: 08 Jan 2023 22:38:29

Reviewer 2 Report

Dear authors,

This is an interesting work and can be published in this journal.

 best wishes

Reviewer 3 Report

Specific comments have been made in the text.

Author Response

Dear Reviewer,

Thanks for your suggestions. We are working on it and will send you the revised manuscript shortly.

Kind regards,

Dr. Marcia Toro

Reviewer 4 Report

The article focused on a very interesting idea and the information has many strong points, somehow it needs to be refined and some crucial points need to be clarified. Abstract needs to revise. Add phylogenetic tree scale. Fig. 3 inserted name in the plate need to modify and clear. Strongly recommended to spell out all abbreviations in the text if it first time mentioned in the text. Typological, grammatical, and other errors should be rectified.

Author Response

(The authors gave the same response as above.)
